# Study on Compressive Properties and Dynamic Characteristics of Polypropylene-Fiber-and-Cement-Modified Iron-Ore Tailing under Traffic Load

**DOI:** 10.3390/polym14101995

**Published:** 2022-05-13

**Authors:** Ping Jiang, Yewen Chen, Xinjiang Song, Na Li, Wei Wang, Erlu Wu

**Affiliations:** 1School of Civil Engineering, Shaoxing University, Shaoxing 312000, China; jiangping@usx.edu.cn (P.J.); ccw181898@163.com (Y.C.); lina@usx.edu.cn (N.L.); wellswang@usx.edu.cn (W.W.); 2Key Laboratory of Ministry of Education for Geomechanics and Embankment Engineering, Hohai University, Nanjing 210098, China; 3Anhui and Huaihe River Water Resources Research Institute, Bengbu 233000, China; sxj06@163.com

**Keywords:** iron-ore tailing, polypropylene fiber, unconfined compressive properties, dynamic properties, cumulative plastic strain

## Abstract

Using polypropylene (PP) fiber and cement to modify iron-ore tailing and applying it to road engineering is an effective way to reuse iron-ore tailing. The compressive properties and deformation characteristics of PP-fiber-and-cement-modified iron-ore tailing (FCIT) under traffic load were studied by the unconfined-compressive-strength (UCS) test and the dynamical-triaxial (DT) test. The test results indicated that the UCS and residual strength both increased with increasing PP-fiber content, and tensile and toughness properties were positively correlated with PP-fiber content. Moreover, the dynamic elastic modulus and damping of FCIT both showed a negative linear relationship with cycle time. It can be found from the test results that 0.75% was the best PP-fiber content to modify iron tailing sand in this work. Lastly, a prediction model was developed to describe the relationship between the cumulative plastic strain, PP-fiber content and cycle time, which can effectively capture the evolution law of the cumulative plastic strain with cycle time of FCITs at different PP-fiber contents.

## 1. Introduction

The stacking of iron-ore tailing has a negative impact on the surrounding environment and people. The resource application of iron-ore tailing is an effective way to reduce the break risk of iron tailing dams. Therefore, some investigations have been focused on the resource application of iron-ore tailing. Yuan [1], Suthers [2] and Lv [3] et al. studied the application of iron-ore tailing in the recovery of valuable metal. Yao [4] and Shettim [5] studied the application of iron-ore tailing as aggregate in the preparation of concrete. Luo [6] and Kuranchi [7] studied the use of iron-ore tailing in brick making. In addition, there have also been some studies and applications of iron-ore tailing in cement composites [8] and filling materials [9,10]. As road-engineering construction needs a large amount of subgrade filler, the large-scale exploitation of river sand and natural sand as road-base materials has seriously affected the natural ecological environment. Zeng [11] and Bastosh [12] et al. investigated the mechanical properties of iron-ore tailing and found that iron-ore tailing can be used as subgrade road filler after chemical stabilization, which can replace river sand and natural sand.

At present, iron-ore tailing needs to be modified with materials so that it can be used as subgrade road filler. Quite a few researchers have mainly focused on modifying materials and investigating the mechanical properties of modified iron-ore tailing. Cement is an inorganic cementitious material commonly used to modify soil [13]. Therefore, some scholars modified iron-ore tailing with cement, such as Li et al. [14], who studied the physical and mechanical properties of cement-modified iron-ore tailings (CITP) and found that the cementation of cement can enhance the bonding between iron-ore tailings and improve its compressive strength and shear strength. Liu et al. [15] verified that CITP can meet the specification requirements of highway pavement base through the unconfined-compressive-strength (UCS) test and pointed out that the strength characteristics of CITP and cement soil are similar. Although the addition of cement can improve the UCS of iron-ore tailing, it can be seen from the studies of many scholars on CITP that the failure mode of CITP is mostly brittle failure, which can lead to serious consequences. Therefore, the fiber is added to CITP in order to change the failure mode. Yang et al. [16] used fiber and cement to modify loess and found that the failure mode of modified loess transitions from brittle failure to ductile failure with the increase in fiber content. Jin et al. [17] studied the UCS of iron-ore tailing modified with PP fiber, and found that the PP fiber can indeed improve the UCS, but the content of PP fiber has little effect on the UCS of modified iron-ore tailing. Jiang et al. [18] mixed polypropylene fiber with iron tailing sand for improvement and explained that PP fiber improves the shear performance of iron-ore tailing from the perspective of energy dissipation.

Random discrete fiber reinforcement is an efficient technology to improve the mechanical properties of soil, which can improve the strength and deformation characteristics of soil under static and dynamic loads to a certain extent. The existing studies mainly focus on the static properties of fiber-reinforced soil; only a few researchers have studied the dynamic response characteristics of fiber-reinforced soil, especially the cumulative strain of reinforced soil. Zhang et al. [19] mainly analyzed the development law of cumulative strain with dynamic stress amplitude and vibration time of fiber-reinforced soil and established an exponential model to describe the relationship between the cumulative strain, amplitude and cycle times. Zhao et al. [20] conducted cyclic loading tests on fiber-reinforced sand and derived an empirical formula between the maximum dynamic shear modulus and damping ratio. Wang et al. [21] put forward a functional relationship between the cumulative strain and cycle time of fiber-reinforced soil. Moreover, the fiber content has a significant impact on the dynamic response characteristics of modified soil. For example, Maher et al. [22] found that both the damping ratio and dynamic shear modulus of fiber-reinforced sand increased nearly linearly with the increase in fiber content. Li et al. [23] found that the dynamic elastic modulus of fiber-reinforced soil increases with the increase in fiber content in the dynamic-triaxial experiment. Orakoglu et al. [24] found that the addition of fiber increases the damping ratio and dynamic shear modulus and established a functional relationship between dynamic shear stress and dynamic shear modulus.

It can be found from the existing research that there are few studies on the mechanical properties of FCIT, especially the strength and deformation characteristics under traffic load. That is, a large number of experiments on the mechanical properties still need to be conducted for the application of FCIT in road engineering. To this end, the unconfined-compressive-strength test and dynamic-triaxial test were conducted to investigate the strength and deformation characteristics of FCIT under traffic load in this paper, which can provide the experimental and theoretical support for the application of modified iron-ore tailing in road engineering.

## 2. Materials and Methods

### 2.1. Materials

The iron-ore tailings used in the test were obtained from the Lizhu iron-tailing plant in Shaoxing, China, as shown in Figure 1. The liquid limit and plasticity index of the iron-ore tailing were 36.95% and 29.35, respectively, and the main compounds are shown in Table 1. The P.O. 42.5 cement used in the test was produced by Shaoxing Zhaoshan Building Materials Co., Ltd. in Shaoxing, China, as shown in Table 2. The PP fiber used in the test was produced by Shaoxing Fiber High Tech Co., Ltd. in Shaoxing, China, and had good dispersion with a length of 6 mm and a diameter of 48 µm, as shown in Table 3.

### 2.2. Test Scheme

The natural environment of iron-ore tailing is convenient for the actual comprehensive utilization of subgrade engineering. With reference to the liquid limit of iron tailing sand, the design moisture content was 30%. Considering economic cost and meeting the road-base requirements of mechanical properties, the cement content was 10%, and the contents of PP fiber were 0%, 0.25%, 0.5%, 0.75% and 1%. The water content, cement content, and PP-fiber content were defined as the ratio of the mass of water to the mass of dry iron-ore tailing, the ratio of the mass of cement to the mass of dry iron-ore tailing, and the ratio of the mass of PP fiber to the mass of dry iron-ore tailing, respectively.

UCS is the most commonly used mechanical index in road engineering. Therefore, the UCS test was used to study the influence of fiber content on the unconfined compressive performance of FCIT under static load. Furthermore, the dynamic-triaxial test was performed to investigate the dynamic characteristics of FCIT under cyclic traffic load. For general highway traffic, the low cyclic stress ratio ($\sigma_{d}/\sigma_{3}$) is in the range of 0.1 to 0.75 [25]. $\sigma_{d}$ is taken as the amplitude of dynamic load, and $\sigma_{3}$ is the confining pressure. According to the amplitude of traffic load, $\sigma_{d}$ during the test was set as 75 kPa. Moreover, the frequency, confining pressure and number of cycles were set as 1 Hz, 100 kPa and 1000, respectively, by considering the normal driving density and speed under general road conditions, the lateral pressure on the general subgrade, and the number of cycles on the subgrade under general traffic flow. The test details are indicated in Table 4.

### 2.3. Specimen Preparation

As shown in Figure 2, the required weight of iron-ore tailing, cement and fiber were poured into a mixing drum. First, the mixture was blended until the three ingredients were well combined. Then, corresponding quality of water was weighed and poured into the mixing bucket. After blending, the FCIT mixture was obtained. The FCIT mixture was poured into the specimen mold with a diameter of 39.1 mm and a height of 80 mm in two batches. After each addition, the FCIT mixture was vibrated up and down with the mold in order to vibrate the air bubbles out. After specimen preparation, the specimen needs to sit for an hour to develop the specimen shape. Next, the white bearing specimen was removed using filter papers to wrap both ends of the specimen, the number was marked, and it was placed in water. Finally, the specimens were placed in a standard curing chamber at 20 ± 2 °C with humidity of 95% for 7 d and 28 d.

### 2.4. Test Apparatus

The UCS test applies axial pressure to the specimen with a constant load P. Figure 3a shows the stress state of UCS test specimen. The instrument used for the UCS test was a TKA-WCY-1F full-automatic multifunctional unconfined-compressive-strength tester, as seen in Figure 3b. Three parallel specimens were used for the test, and the test loading rate was 1 mm/min.

The DT test is a commonly used method to test the dynamic characteristics of soil. The stress state of the DT-test specimen is indicated in Figure 4a. The instrument used in the DT test was a GDS dynamic-triaxial instrument, as shown in Figure 4b. The dynamic load with the form of a sine wave was adopted in the DT test and 20 data points were collected in each cycle.

## 3. Results and Discussion

### 3.1. Unconfined Compressive Properties

#### 3.1.1. Characteristics of Stress–Strain Curve

Figure 5 gives the stress–strain curves of FCITs with different PP-fiber contents at the curing ages of 7 d and 28 d. It can be seen from Figure 5 that there are four stages in the stress–strain curve of FCIT: the linear-elastic stage, plastic stage, stress-attenuation stage and residual-strength stage. Moreover, with the increase in PP-fiber content, the brittleness of the specimen decreases and the plasticity increases, indicating that the PP fiber has an obvious strengthening effect on the plasticity of the cement-stabilized iron tailing.

In the UCS-test process, the deformation of the specimen is effectively weakened due to the friction and tensile action of the PP fiber. The bonding action between the PP fiber and the soil particle inhibits the relative sliding of the soil particles and further delays the cracking of the specimen. The bonding force between the PP fiber and soil interface increases with increasing PP-fiber content, and a spatial-network structure is formed between staggered PP fibers, which can restrict the sliding and rolling of soil particles. That is, the stability of the specimen is improved at the spatial level and the plasticity is increased.

#### 3.1.2. UCS and Residual Strength

From the stress–strain curves in Figure 5, the UCSs of FCITs with different PP-fiber contents and curing ages can be obtained, as indicated in Figure 6. The UCSs of FCITs with different PP-fiber contents at the age of 7 d are greater than 1 MPa which meets the strength requirement of relevant specification [26] for cement-stabilized material. As indicated in Figure 6, the UCSs of FCITs with different PP-fiber contents all increase by more than 35% with increasing curing age. The reason for the increasing strength is the increased hydration products of cement, such as C-S-H and C-A-S-H. On the one hand, these hydration products fill the pores between particles. On the other hand, the bonding between particles is strengthened through cementation, as well as the bonding between particles and PP fibers. Therefore, the addition of PP fiber can improve the UCS of FCIT. Taking the specimens with the curing age of 7 d as an example, compared with the specimen without fiber, the UCSs of the specimens with PP-fiber contents of 0.25%, 0.5%, 0.75% and 1% are increased by 5%, 13%, 20% and 17%, respectively. The UCS of the specimen with the curing age of 28 d has a similar change law as the specimen with the curing age of 7 d. The addition of PP fiber improves the ability of the specimen to resist external force, so as to improve the strength of the specimen. The UCS first increases and then decreases with the increasing content of PP fiber. The reason for this change is that the fiber will present a spatial-network structure in the specimen, and the agglomeration phenomenon will occur in the specimen with undue PP fiber, which will cause the cement particles and iron-ore-tailing particles to be unable to fully contact, thereby decreasing the UCS.

Figure 7 shows the relationship between the residual strength and the PP-fiber content of the FCIT specimen. It can be seen from Figure 7 that the PP-fiber contents of 0.25% and 0.5% have little reinforcement effects on the residual strength of FCIT. On the contrary, there is an obvious improvement effect on the residual strength of FCIT with the PP-fiber content of 0.75%. Compared to fiberless samples, the residual strength of the samples with 0.75% fiber content increases by 320% and 580% at 7 d and 28 d, respectively. Significantly, when the PP-fiber content increases from 0.75% to 1%, there is little improvement effect on the residual strength of FCIT, especially the residual strength at 28 d. In fact, the PP fibers in the specimen can form a staggered spatial-network structure, and this network structure restricts the slip of specimen particles so that the deformation resistance of the specimen is enhanced. Especially, it can be seen from Figure 6 and Figure 7 that the reinforcement effect is best when the PP-fiber content is 0.75%.

#### 3.1.3. Brittleness Index, Toughness Index, and Modulus Strength Ratio

Brittle materials often fail suddenly without warning when impacted by external forces. Although cement can better improve the strength of iron-ore tailing, its brittleness is also greatly increased [27]. In order to evaluate the effect of PP-fiber content on the brittle failure of FCIT, the brittleness index ($I_{B}$) is introduced to quantitatively describe the failure mode of FCIT. The value of $I_{B}$ can be calculated according to Equation (1) [28]. The calculated values of $I_{B}$ are indicated in Figure 8. As shown in Figure 8, added PP fiber can effectively reduce the brittleness index of FCIT and gradually change its failure mode from brittleness to plasticity, so as to improve the safety of the pavement base. Obviously, the value of the brittleness index linearly decreases with the increase in PP-fiber content, and the PP-fiber content and brittleness index of FCIT meet the linear relationship, as shown in Equation (2).
(1)$$I_{B}=\frac{q_{u}-q_{c}}{q_{u}}$$
where $q_{u}$ is the unconfined compressive strength, and $q_{c}$ is the residual strength.
(2)$$I_{B}=I_{B0}-\Delta I_{B}f$$
where *f* is PP-fiber content, $I_{B0}$ is the brittleness index when *f* = 0, and $\Delta I_{B}$ is a material parameter and reflects the decreasing rate of $I_{B}$ with increasing *f*.

In order to describe the mechanical properties of FCIT after yield, the energy dissipation of FCIT after yield can be quantitatively characterized by the toughness index ($T_{I}$). The higher the toughness index, the better the ability of the absorbing energy after yield [29]. $T_{I}$ can be calculated according to Equation (3), and then the calculated values of $T_{I}$ are shown in Figure 9. Apparently, there is an exponential relationship between the content of PP fiber and $T_{I}$, as shown in Equation (4). It can be found that there is an increase in the difference between $T_{I}$ of 7 d and 28 d with increasing PP-fiber content. The reason for this is the more sufficient hydration reaction of cement and the more obvious interfacial bonding between the PP fiber and iron-ore-tailing particles so that FCIT can absorb more strain energy after yield.
(3)$$T_{I}=\frac{\int_{\varepsilon_{q}}^{\varepsilon_{q}+5\%}\sigma (\varepsilon )d\varepsilon }{5\%}$$
where, $T_{I}$ is the strain corresponding to the maximum value of stress.
(4)$$T_{I}=T_{I0}e^{\Delta T_{I}f}$$
where $T_{I0}$ is the toughness index when *f* = 0, and $\Delta T_{I}$ is a material parameter.

Due to the tensile action of PP fiber, the crack expands slowly in the specimen after the FCIT specimen yields. The broken test block does not immediately fall but is bonded to the specimen (Figure 10), which can continue to bear the load. That is, the broken test block can absorb part of the external energy and improve the toughness index of FCIT.

The modulus strength ratio can be used to characterize the tensile performance of FCIT. The modulus strength ratio refers to the ratio of soil modulus to strength [30,31], as given by
(5)$$\eta =\frac{E_{50}}{q_{u}}$$
where, $E_{50}$ is the secant modulus.

There is a negative correlation between the modulus strength ratio and the tensile performance of the material. The modulus strength ratio of FCIT can be calculated through Equation (5), as shown in Figure 11. It can be seen from Figure 11 that the modulus strength ratio of FCIT decreases with the increase in fiber content, indicating that the addition of fiber can improve the tensile properties of FCIT.

## 4. Dynamic Characteristics

When the pavement base is subjected to cyclic traffic load, the base material will produce strain accumulation, and the cumulative strain linearly increases with the increasing number of cycles. Finally, it will evolve into two possible results. One is a strain-softening type in which the soil deformation reaches the limit value and then fails. The other is a strain-hardening type in which the material strength increases with increasing strain and the dynamic equilibrium is achieved when the final strain reaches a certain value. Dynamic stress–strain characteristics, elastic modulus, damping ratio and cumulative strain can be used to characterize the dynamic characteristics of FCIT under cyclic traffic load.

### 4.1. Dynamic Stress–Strain Characteristics

Figure 12 shows the stress–strain hysteretic curve of FCIT under cyclic loading. The red dotted arrow in Figure 12 indicates the test loading process. A complete cycle is formed by applying compressive stress to the specimen and releasing the stress. The hysteresis-loop area can effectively reflect the damping and energy dissipation of soil. The fuller the hysteresis loop in Figure 12, the greater the damping ratio of the specimen, that is, the greater the energy dissipation. According to the hysteresis loop in the figure, the dynamic elastic modulus *E* and damping ratio λ can be calculated.
(6)$$E=\frac{L_{\mathrm{AB}}}{L_{\mathrm{OB}}}$$
(7)$$\lambda =\frac{S_{h}}{4\pi S_{a}}$$
where $L_{\mathrm{AB}}$ is the length of AB, and $L_{0B}$ is the length of OB; $S_{h}$ represents the area of hysteresis loop, that is, the energy loss of loading one cycle, and $S_{a}$ represents the triangular AOB area.

The stress–strain hysteretic curves of FCIT are shown in Figure 13. It can be seen from the figure that the hysteretic circle moves in the direction of increasing strain with the increasing number of cycles N. Obviously, there is an accumulation of the strain of FCIT. At the same time, it can be determined from Figure 14 that the distance between hysteretic loops gradually decreases with the increasing number of cycles, which indicates that the cumulative strain rate gradually decreases and eventually tends to zero with the increasing number of cycles.

Figure 15 shows the hysteresis loops (*N* = 500) of FCITs with different PP-fiber contents. It is found from Figure 15 that PP-fiber content has an obvious impact on the dynamic elastic modulus and damping ratio of FCIT.

### 4.2. Evolution Law of Dynamic Characteristics with Cycle Times

#### 4.2.1. Dynamic Elastic Modulus

The dynamic elastic modulus is an important mechanical index of pavement base. The computed values of dynamic elastic modulus are plotted against cycle times (see Figure 16). As illustrated in Figure 16, the dynamic elastic modulus decreases with the increase in the number of cycles, and there exists a linear relationship between the dynamic elastic modulus and the number of cycles. It can also be found that the dynamic elastic modulus of FCIT without PP fiber decreases faster. The reason for this is that the fiberless specimen is brittle material, and the internal damage of the specimen without the reinforcement effect of PP fiber develops faster when subjected to external force. That is, the addition of PP fiber can reduce the internal-damage rate of the specimen. Additionally, it can be seen that the dynamic elastic modulus of the specimen with PP fiber first increases and then decreases with increasing PP-fiber content. Obviously, the dynamic elastic modulus of the specimen with the PP-fiber content of 0.75% is the greatest among the specimens with PP fiber, which is similar to the change law of UCS.

#### 4.2.2. Damping Ratio

The damping ratio is used to describe the speed of energy dissipation of the pavement base in the process of traffic-load vibration. Figure 17 gives the test data between the damping ratios and cycle times of specimens with different PP-fiber contents. It can be seen that there is a linear relationship between the damping ratio and the number of cycles. Especially, added PP fiber can effectively improve the damping ratio of FCIT. This is because the addition of fiber allows the specimen to absorb more energy during vibration and promote vibration attenuation.

### 4.3. Development Law of Cumulative Plastic Strain with the Number of Cycles

#### 4.3.1. Porosity

The porosities of FCIT specimens with different PP-fiber contents are different, and the porosity has a significant impact on the dynamic characteristics of FCIT. The volume of the specimen is supposed as 1, and the porosity *n* can be calculated based on Equation (8).
(8)$$\left\{ \begin{array}{l} V_{s}=\frac{m_{s}}{G_{s}}=\frac{\rho_{d}}{(1+f+c)G_{s}}\\ V_{f}=\frac{m_{f}f}{G_{f}}\\ V_{c}=\frac{m_{c}c}{G_{c}}\\ n=1-V_{s}-V_{f}-V_{c} \end{array}\right.\;$$
where $V_{s}$, $V_{f}$ and $V_{c}$ are the volumes of iron-ore tailing sand, PP fiber and cement, respectively; $m_{s}$, $m_{f}$ and $m_{c}$ are the qualities of iron-ore tailing sand, PP fiber and cement, respectively; $G_{s}$, $G_{f}$ and $G_{c}$ are the proportions of iron-ore tailing sand, PP fiber and cement, respectively; *c* is the cement content, *f* is the PP-fiber content, and $\rho_{d}$ is the dry density of specimen.

The porosity of FCIT before the test are then calculated according to Equation (8), as shown in Figure 18. It is intuitively found from Figure 18 that the porosity increases with the increase in PP-fiber content. The reason for this is that the PP fibers are randomly distributed in the specimen, and the PP fibers are in the state of intersection and interleaving, which increases the porosity of the specimen.

#### 4.3.2. Analysis of Cumulative Deformation Characteristics

Cumulative plastic strain refers to the plastic strain produced in the whole process of loading–unloading–loading, and it is suitable for evaluating the internal-damage evolution degree of FCIT under traffic load. Figure 19 shows the cumulative plastic strain $\varepsilon_{p}$, elastic strain $\varepsilon_{e}$ and total strain $\varepsilon_{d}$. The total strain is composed of cumulative plastic strain and elastic strain [32]. The cumulative plastic strain is shown in the red curve in Figure 19, which is the valley value in each sine wave.

The test results show that there is an accumulation of the deformation of FCIT with increasing cycle time. Figure 20 shows the test data between the cumulative plastic strain and cycle time of FCIT at different PP-fiber contents. From Figure 20, it can be seen that the cumulative plastic strain increases at a decreasing rate with increasing cycle time and increases with the increase in PP-fiber content. As previously discussed, more fibers can cause more pores in the specimen, and this is the reason for the cumulative plastic strain increasing with increasing PP-fiber content. That is, there is a positive relation between the cumulative plastic strain and PP-fiber content.

From Figure 20, it is apparent that the strain curve is the strain-hardening type. That is, the cumulative plastic strain increases at a decreasing rate with increasing cycle time, and eventually tends to be constant. However, the semi-logarithmic function is used to describe the relationship between the cumulative plastic strain and cycle time [33,34], in which the cumulative plastic strain will increase with increasing cycle time. For this reason, the hyperbolic function is adopted to describe the relationship between the cumulative plastic strain and cycle time, as given by
(9)$$\varepsilon_{p}=\frac{AN}{B+N}$$
where *A* and *B* are the two material parameters, and *A* is regarded as the final cumulative plastic strain.

Equation (9) was used to fit the test data between the cumulative plastic strain and cycle time, as illustrated in Figure 20. As presented in Figure 20, the coefficients of determination *R*^2^ are all greater than 0.980, which indicates that Equation (9) can effectively express the relationship between the cumulative plastic strain and cycle time.

Given that there is a positive relation between the cumulative plastic strain and PP-fiber content, the values of cumulative plastic strain were then plotted against the PP-fiber content at cycle times of 200, 600 and 1000, as shown in Figure 21. It is illustrated in Figure 21 that an exponential relationship exists between the cumulative plastic strain and PP-fiber content, as given by
(10)$$\varepsilon_{p}=\exp(\alpha +\beta f^{1.5})$$
where α and β are the two material parameters.

Equation (10) was used to fit the test data between the cumulative plastic strain and PP-fiber content (Figure 21). Obviously, a good agreement can be found between the test data and curves fitted by Equation (10) from Figure 21. Based on Equations (9) and (10), a prediction model for the cumulative plastic strain considering cycle time and PP-fiber content can be expressed as
(11)$$\varepsilon_{p}=\exp(\alpha +\beta f^{1.5})\frac{N}{N+\omega }$$
where ω are the two material parameters

In order to verify the validity of the proposed prediction model, Equation (11) was used to simulate the deformation behaviors of FCIT at different cycle times and PP-fiber contents, as presented in Figure 22. The coefficient of determination *R*^2^ fitted by Equation (11) is 0.981, which indicates that good effects in simulation have been achieved. At the same time, it can be found that there is a satisfactory agreement between the test data and model simulations. As observed, the proposed model can effectively capture the evolution law of the cumulative plastic strain with increasing cycle time at different PP-fiber contents of FCIT.

## 5. Conclusions

Modifying iron-ore tailing with PP fiber and cement can improve its mechanical properties so that modified iron-ore tailing can be applied in road engineering, which realizes the reuse of iron-ore tailing. This paper investigated the strength and deformation behaviors of modified iron-ore tailing through the unconfined-compressive-strength test and the dynamic-triaxial test, which aims to provide a theoretical basis for the application of modified iron-ore tailing in road engineering. The main conclusions are as follows:(1)The addition of PP fiber increases the UCS and residual strength of FCIT, and 0.75% has the best effect on improving the UCS and residual strength among all PP-fiber contents.(2)The brittleness index and modulus strength ratio both decrease with increasing PP-fiber content, and the addition of PP fiber efficiently improves the tensile and toughness properties of FCIT.(3)The dynamic elastic modulus and damping ratio of FCIT both meet the linear relationship with cycle time. The dynamic elastic modulus of FCIT first increases and then decreases, and the damping ratio first decreases, then increases, and finally decreases among specimens with increasing PP-fiber content from 0.25% to 1%.(4)The deformation characteristics of FCIT under cycle load are related to the PP-fiber content, and the cumulative plastic strain increases with increasing PP-fiber content.(5)A prediction model is developed for simulating the deformation behaviors of the dynamic-triaxial test, which can effectively capture the evolution law of the cumulative plastic strain with cycle time of FCIT at different PP-fiber contents.

## Figures and Tables

**Figure 1 polymers-14-01995-f001:**
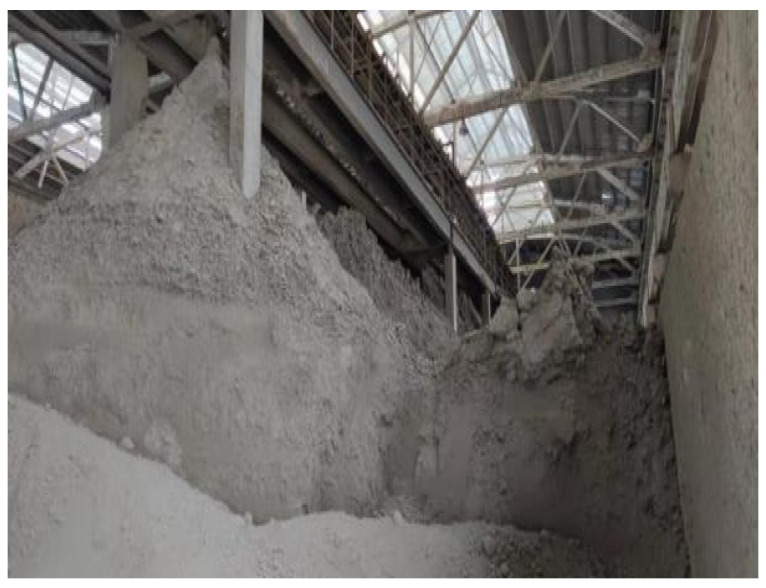
Iron-ore tailings for test.

**Figure 2 polymers-14-01995-f002:**
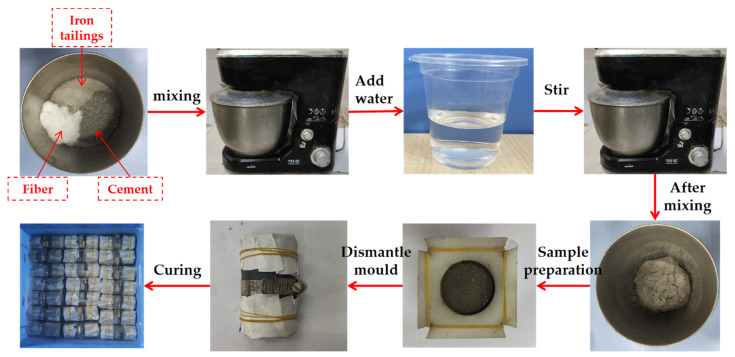
Sample preparation flow.

**Figure 3 polymers-14-01995-f003:**
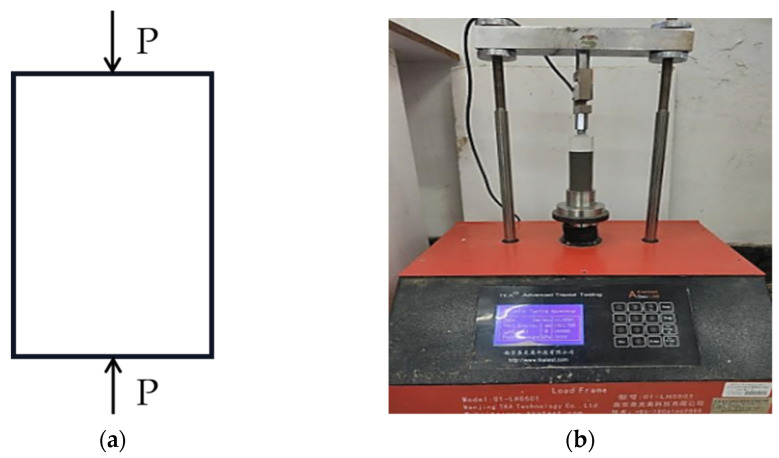
UCS test. (**a**) Unconfined-stress state of specimen; (**b**) Test apparatus.

**Figure 4 polymers-14-01995-f004:**
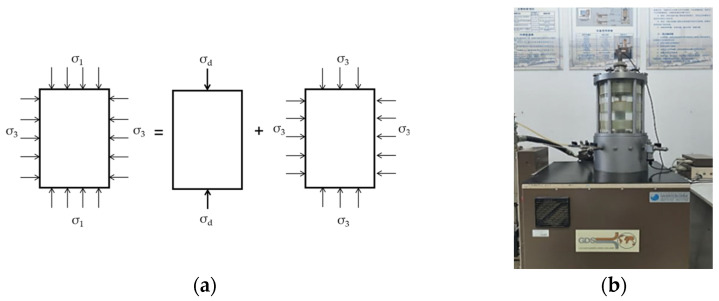
DT test. (**a**) Triaxial-stress state of specimen; (**b**) Test apparatus.

**Figure 5 polymers-14-01995-f005:**
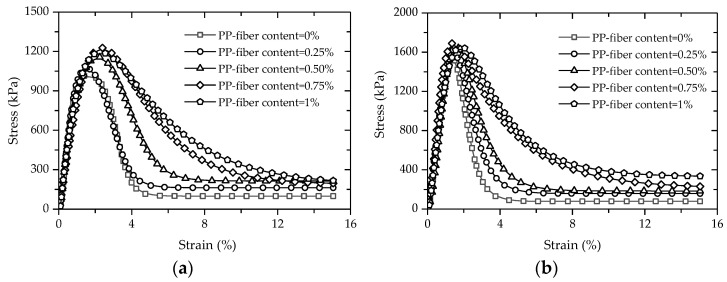
Stress–strain behaviors of specimen in unconfined-compressive test: (**a**) 7 d; (**b**) 28 d.

**Figure 6 polymers-14-01995-f006:**
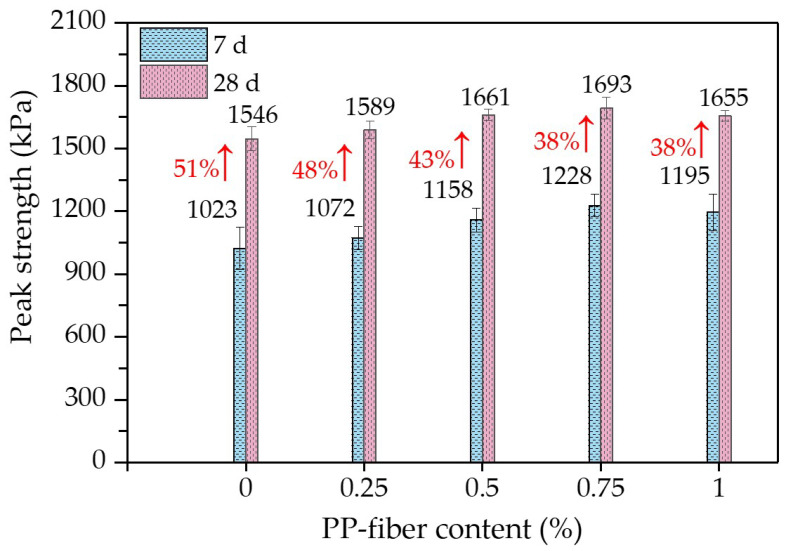
Unconfined compressive strengths of FCIT at different ages.

**Figure 7 polymers-14-01995-f007:**
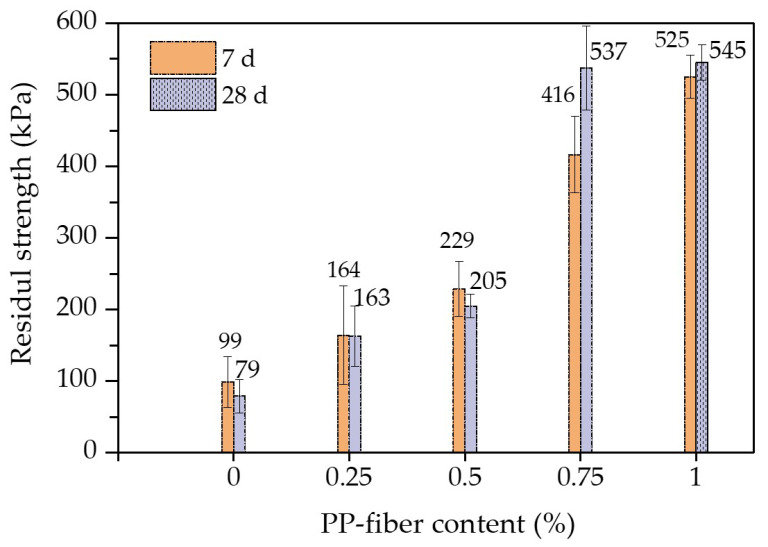
Residual strengths of FCIT at different curing ages.

**Figure 8 polymers-14-01995-f008:**
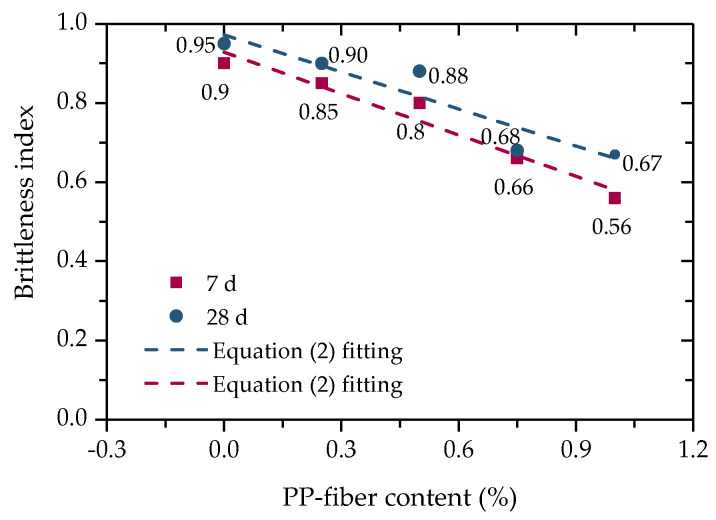
Relationship between brittleness index and PP–fiber content.

**Figure 9 polymers-14-01995-f009:**
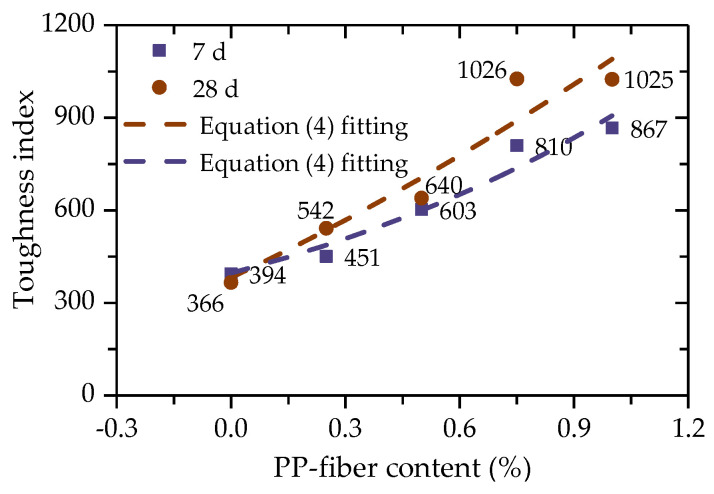
Relationship between toughness index and PP–fiber content.

**Figure 10 polymers-14-01995-f010:**
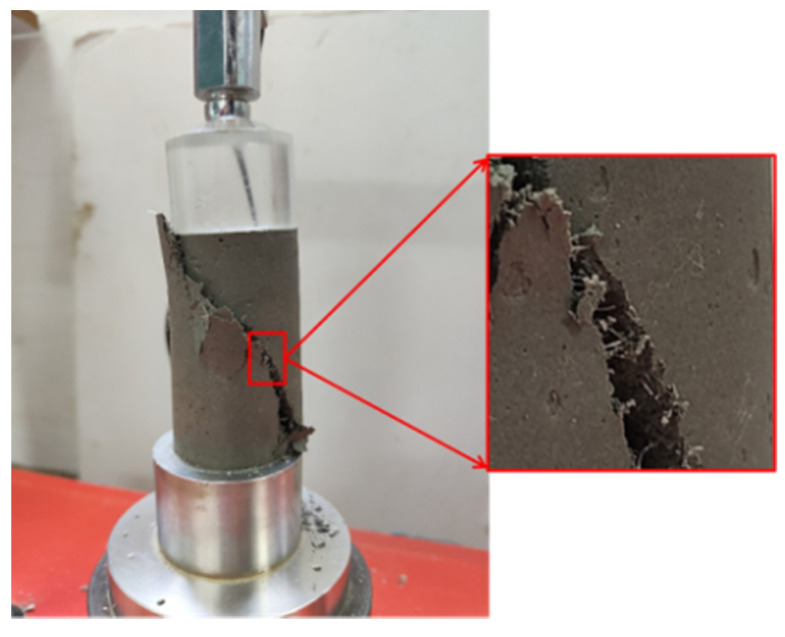
Damaged surface of specimen.

**Figure 11 polymers-14-01995-f011:**
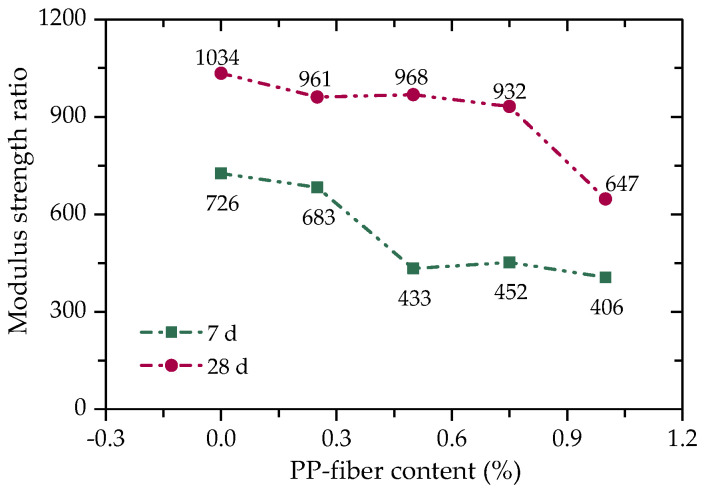
Relationship between modulus strength ratio and PP-fiber content.

**Figure 12 polymers-14-01995-f012:**
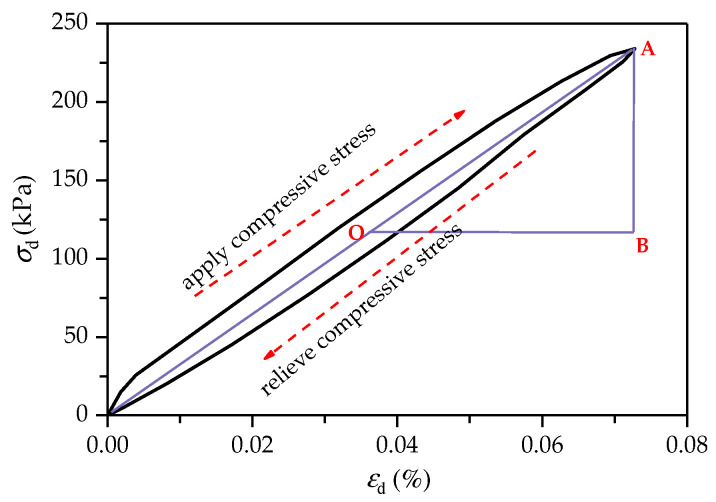
Stress–strain hysteretic curve of FCIT under cyclic loading.

**Figure 13 polymers-14-01995-f013:**
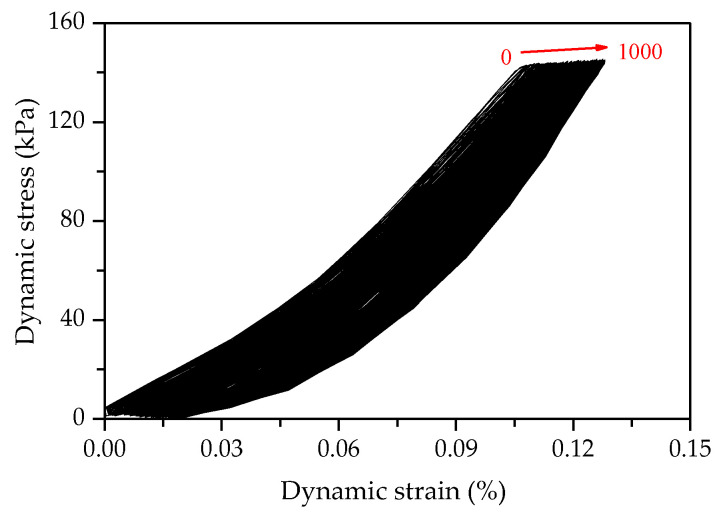
Hysteresis loop.

**Figure 14 polymers-14-01995-f014:**
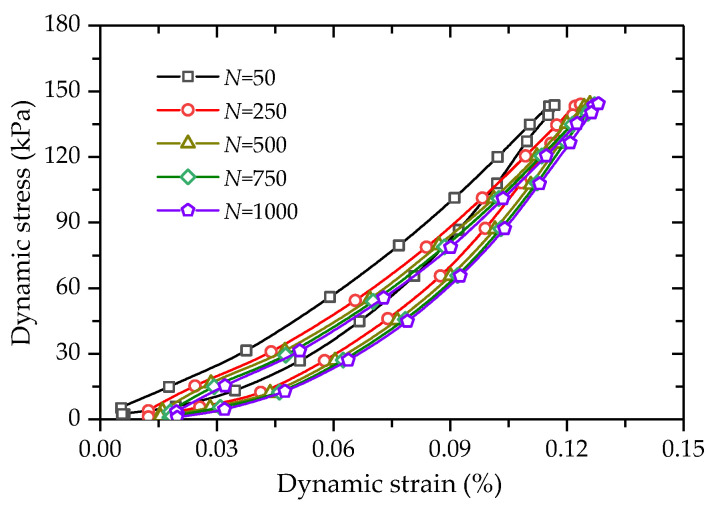
Variation of hysteresis loop with increasing cycle time.

**Figure 15 polymers-14-01995-f015:**
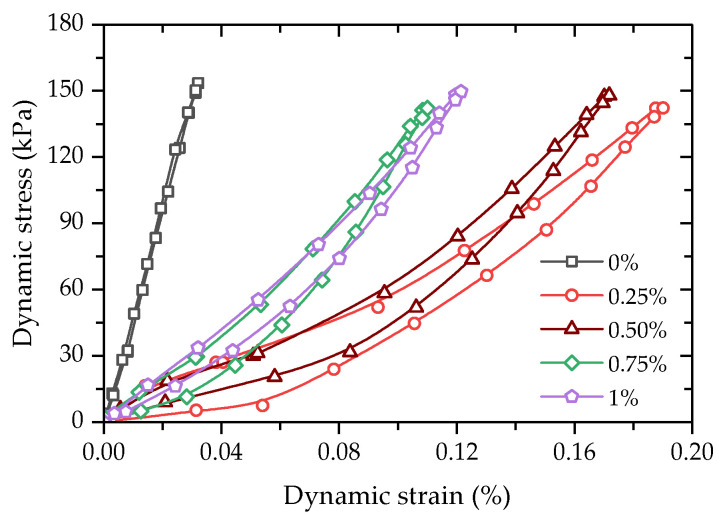
Hysteresis loops of FCIT at different PP-fiber contents (*N* = 500).

**Figure 16 polymers-14-01995-f016:**
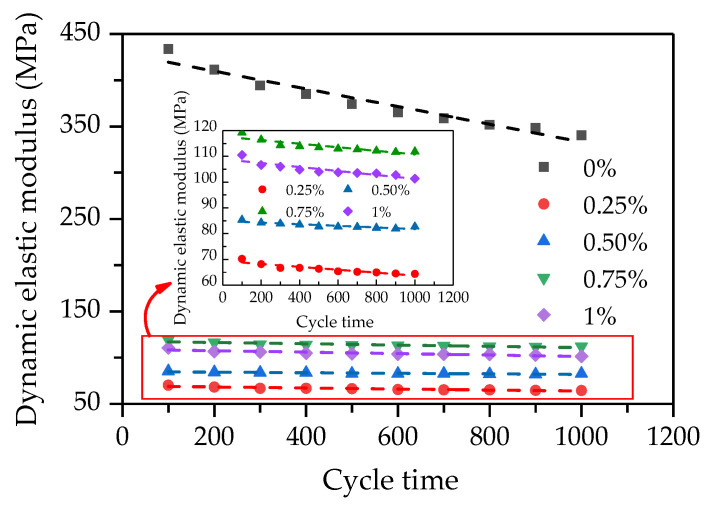
Relationship between dynamic elastic modulus and cycle time.

**Figure 17 polymers-14-01995-f017:**
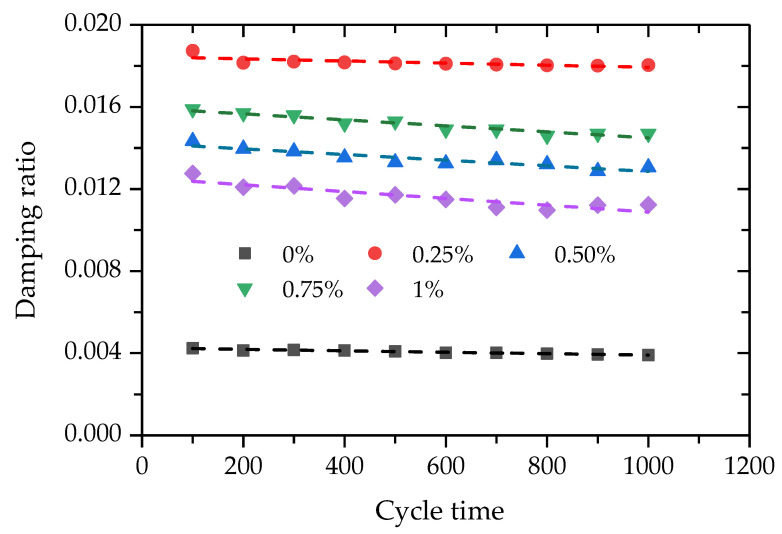
Relationship between damping ratio and cycle time.

**Figure 18 polymers-14-01995-f018:**
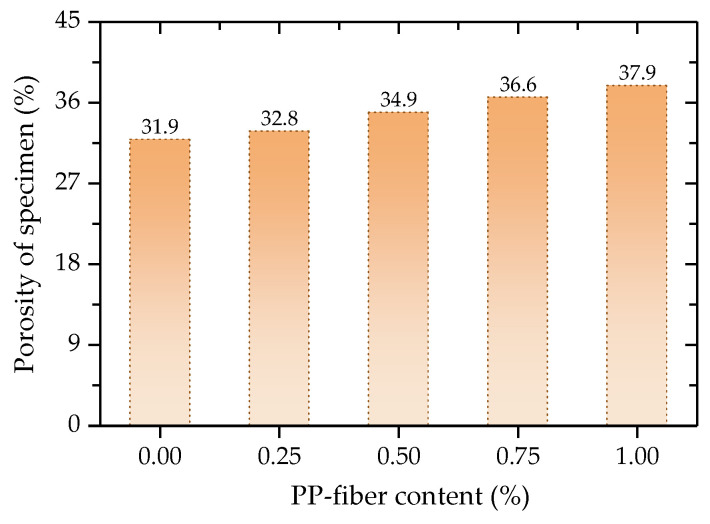
Porosity of specimen at different fiber contents.

**Figure 19 polymers-14-01995-f019:**
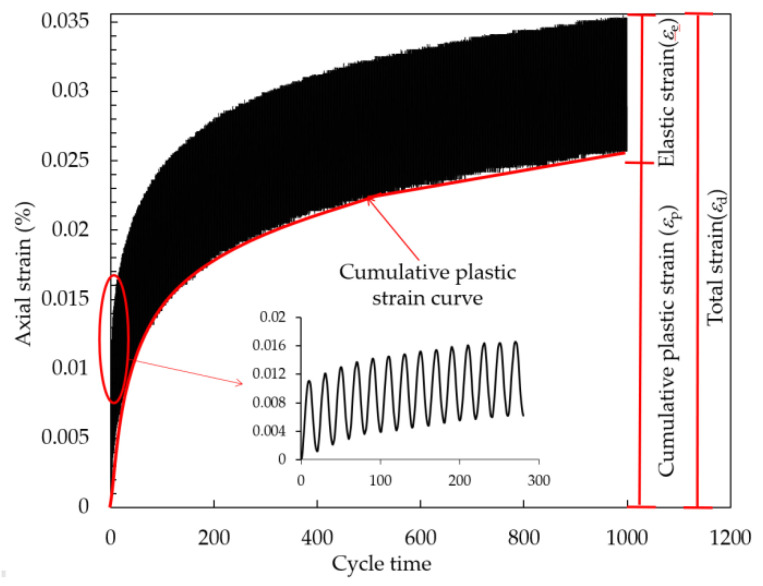
Strain curves with increasing cycle time.

**Figure 20 polymers-14-01995-f020:**
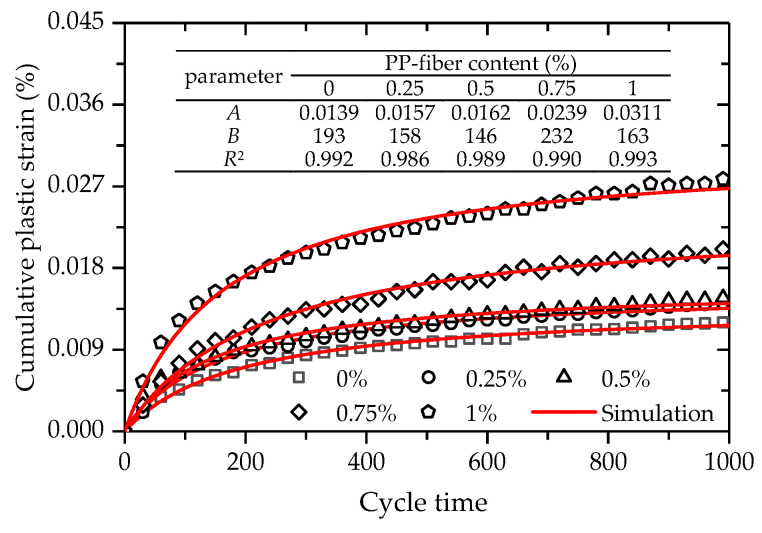
Relationship between the cumulative plastic strain and cycle time.

**Figure 21 polymers-14-01995-f021:**
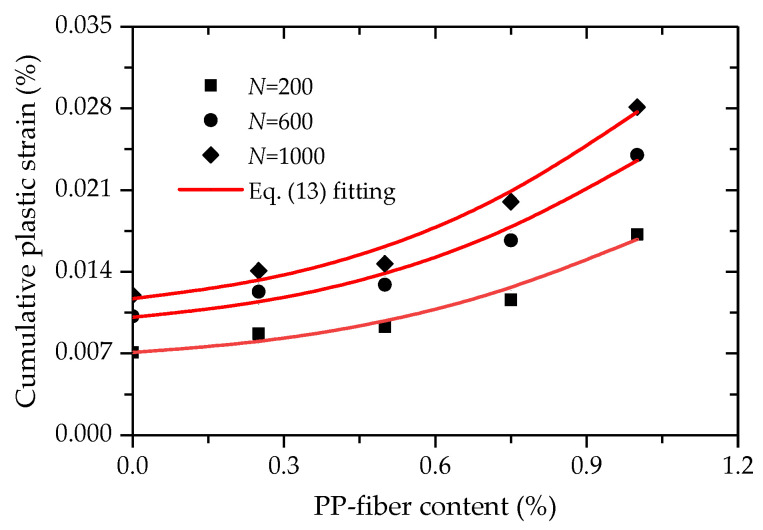
Relationship between the cumulative plastic strain and PP-fiber content.

**Figure 22 polymers-14-01995-f022:**
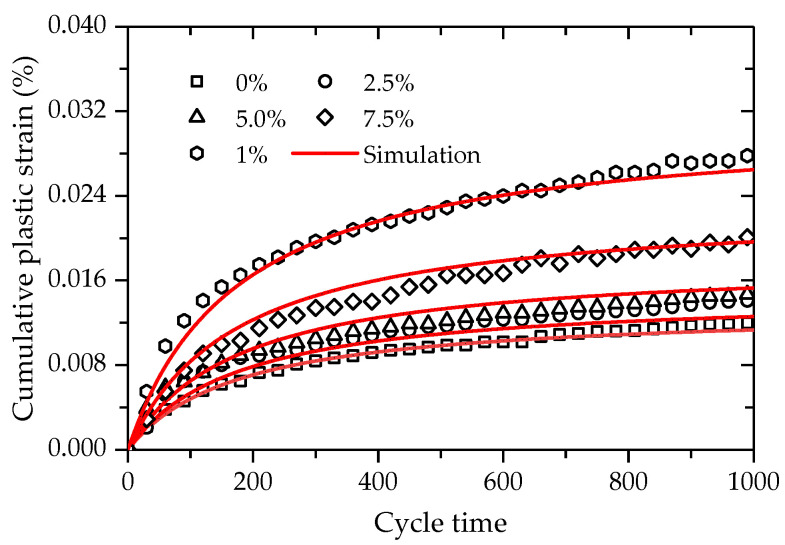
Simulations of the cumulative plastic strain.

**Table 1 polymers-14-01995-t001:** Main chemical composition and physical properties of iron-ore tailing.

Composition (%)			Liquid Limit (%)	Plasticity Index (%)
CaO	Fe_2_O_3_	SiO_2_		
24.8	22.9	21.2	36.95	29.35

**Table 2 polymers-14-01995-t002:** Physical and mechanical properties of P.O. 42.5 cement.

Initial Setting Time (Min)	Final Setting Time (Min)	Compressive Strength		Break Off Strength	
		3 d	28 d	3 d	28 d
≥45	≤600	≥17	≥42.5	≥3.5	≥5.5

**Table 3 polymers-14-01995-t003:** Physical properties of polypropylene fiber.

Diameter (mm)	Specific Gravity (g/cm^3^)	Tensile Strength (MPa)	Elastic Modulus (MPa)
0.048	0.91	340	4200

**Table 4 polymers-14-01995-t004:** FCIT test scheme.

Test	PP Content (%)	Loading Rate (mm/min)	Loading Frequency (Hz)	Cycle Time	Amplitude (kPa)	Confining Pressure (kPa)
UCS	0	1	—	—	—	0
	0.25					
	0.50					
	0.75					
	1					
DT	0	—	1	1000	75	100
	0.25					
	0.50					
	0.75					
	1					

## Data Availability

Not applicable.

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
