# Peer review of "Study on Compressive Properties and Dynamic Characteristics of Polypropylene-Fiber-and-Cement-Modified Iron-Ore Tailing under Traffic Load"

_polymers, 2022, doi:10.3390/polym14101995_

Round 1

Reviewer 1 Report

The paper investigates to use PP fiber and cement to modify iron ore tailing and applying modified iron ore tailing to road engineering. There are following questions:

  1. Abstract and conclusion need to be rewritten to report about the actual quantitative findings of this study are not general descriptions.
  2. The author should compare the results with others’ research to confirm the contribution of this research.
  3. Please supplement the basic physical and chemical properties of the research materials.
  4. Please describe the criteria for determining the test parameters in Table 1.
  5. Please describe the proportions of this study.
  6. It needs to be clearly stated the contributions of the manuscript in the introduction section.
  7. The authors should provide the mechanism and theories for explanations.

Author Response

We would like to thank the reviewers for their valuable comments, which certainly help to improve quality of our manuscript. We have addressed and revised our manuscript accordingly, and the revised parts in the manuscript have been marked in red. Our reply to reviewers’ comments on a point-by-point basis is listed as follows: (Letters C & R denote Comment and Response, respectively).

C1.    Abstract and conclusion need to be rewritten to report about the actual quantitative findings of this study are not general descriptions.

R1.    Thank you for your comment. The abstract and conclusion haven been rewritten in the revised manuscript.

C2.    The author should compare the results with others’ research to confirm thecontribution of this research.

R2.    Thanks. An important conclusion in this paper is the prediction model which is developed to describe the relationship between the cumulative plastic strain, PP fiber content and cycle time. It can be seen from the exist references that the relationship between the cumulative strain and cycle time is described by the logarithmic function. In this paper, hyperbolic function is used to describe the relationship between the cumulative strain and cycle time, because the test results show that the cumulative strain increases at a decreasing rate with cycle time increasing. That is, the cumulative strain will tend to be constant finally. Compared with logarithmic function, the hyperbolic function can well reflect this feature.

C3.    Please supplement the basic physical and chemical properties of the research materials.

R3.    Thank you for your suggestion. The chemical composition and physical properties of the research material has been added in the revised manuscript, as Tables 1, 2 and 3 show.

C4.  Please describe the criteria for determining the test parameters in Table 1.

R4.    Your comments are very helpful to us, and we have not explained how to determine test parameters. Therefore, supplementary explanations are given in the introduction of the test scheme. According to the driving conditions on general highways, the basic parameters and conditions of the test are determined. Specific modifications are:

The frequency is 1 Hz, which represents the normal driving density and speed under general road conditions. The confining pressure is 100 kPa, which represents the lateral pressure on the general subgrade. The number of cycles is 1000, which represents the number of cycles on the subgrade under general traffic flow.

C5.    Please describe the proportions of this study.

R5.    Thank you very much for your comment. The proportion of iron tailing, fiber, cement and water is clearly explained in the first paragraph of the 2.2 section.

C6.    It needs to be clearly stated the contributions of the manuscript in the introduction section.

R6.    Thanks for your suggestion. In the introduction section, the contributions of the manuscript have been clearly indicated.

C7.    The authors should provide the mechanism and theories for explanations.

R7.   Thank you very much for your comments on our manuscript. The experimental results are explained in many places in this manuscript, and the role of fiber in the sample and the influence of this role on the experimental results are explained in detail. 

Reviewer 2 Report

The state-of-the-art report, from one side, is made in details. However, a critical analysis is absent. For this reason, the purpose and tasks of the research are not well formulated and do not reflect real scientific tasks to be solved in order to improve the situation. The study on compressive properties and dynamic characteristics is an engineer task with practical value.

Author Response

We would like to thank the reviewers for their valuable comments, which certainly help to improve quality of our manuscript. We have addressed and revised our manuscript accordingly, and the revised parts in the manuscript have been marked in red. Our reply to reviewers’ comments on a point-by-point basis is listed as follows: (Letters C & R denote Comment and Response, respectively).

  1. The state-of-the-art report, from one side, is made in details. However, a critical analysis is absent. For this reason, the purpose and tasks of the research are not well formulated and do not reflect real scientific tasks to be solved in order to improve the situation. The study on compressive properties and dynamic characteristics is an engineer task with practical value.
  2. Thank you very much for your comments on this work. In view of the innovation and purpose of the work, we further made a supplementary explanation in the introduction and conclusion.